# Evaluating the Effectiveness of Ultrasound-Guided Subacromial-Subdeltoid Bursa and Coracohumeral Ligament Corticosteroid Injections With and Without Physiotherapy in Adhesive Capsulitis Treatment

**DOI:** 10.3390/biomedicines12122668

**Published:** 2024-11-22

**Authors:** Chu-Wen Tang, Ting-Yu Lin, Peng-Chieh Shen, Fuk-Tan Tang

**Affiliations:** 1Department of Physical Medicine and Rehabilitation, Fu Jen Catholic University Hospital, New Taipei City 24352, Taiwan; 2Department of Physical Medicine and Rehabilitation, Lotung Poh-Ai Hospital, Lo-Hsu Medical Foundation, Inc., Yilan 26546, Taiwan; t840326@icloud.com (T.-Y.L.); jamesshen98360315@gmail.com (P.-C.S.); 3Department of Physical Medicine and Rehabilitation, Linkou Chang Gung Memorial Hospital, Taoyuan 33305, Taiwan

**Keywords:** ultrasound-guided injection, subacromial-subdeltoid bursa, coracohumeral ligament, frozen shoulder, shoulder pain

## Abstract

**Background**: The objective of this study was to investigate the effect of ultrasound-guided corticosteroid injection to the subacromial-subdeltoid bursa (SSB) and coracohumeral ligament (CHL) in treating adhesive capsulitis, with a particular focus on evaluating the potential benefits of regular electrotherapy and conventional rehabilitation exercises. **Methods**: A total of 29 patients with unilateral shoulder pain and restricted shoulder range of motion (ROM) were included. Corticosteroids were delivered to the subacromial-subdeltoid bursa (SSB) and coracohumeral ligament (CHL) through a single percutaneous injection. Group 1 consisted of 18 patients who received injections only, while Group 2 comprised 11 patients who received injections in combination with regular physiotherapy. Shoulder Pain and Disability Index (SPADI) scores and ROM were assessed before the injection, and again at 4, 8, and 12 weeks following the intervention. A multivariate mixed-effects model with repeated measurements was conducted for the variables. **Results**: Electrotherapy and traditional rehabilitation exercises did not enhance the effectiveness of this injection approach. Pain, upper extremity function, and ROM in all planes were all significantly improved with a corticosteroid injection to the CHL and SSB. **Conclusions**: Physiotherapy did not offer additional benefits when combined with ultrasound-guided corticosteroid injection to the CHL and SSB. The injection alone significantly improved pain, disability, and ROM in patients with adhesive capsulitis. Further research is required to optimize current physiotherapy with electrotherapy and traditional rehabilitation exercises after ultrasound-guided corticosteroid injections.

## 1. Introduction

Shoulder pain is the third most frequently encountered musculoskeletal complaint in clinics and imposes a significant burden on patients’ daily activities as well as healthcare costs [1]. Adhesive capsulitis is a common shoulder disorder with a lifetime prevalence of 2–5%, particularly affecting middle-aged women [2,3]. Commonly referred to as “frozen shoulder”, the condition is characterized by pain and progressive, global glenohumeral joint range of motion (ROM) limitation [3]. Although adhesive capsulitis is generally considered a self-limiting disease that resolves within two to three years, symptoms can be debilitating and may persist in up to 50% of patients [4].

Pathogenesis of adhesive capsulitis is complex and likely multifactorial. Chronic inflammation, fibroblast proliferation, and an imbalance in extracellular matrix turnover ultimately lead to capsular stiffness [4]. Notably, the predominant pathological changes are found at the rotator interval, both radiologically and histologically [5,6,7]. The rotator interval is a triangular anatomical space located at the anterosuperior aspect of the shoulder, bordered superiorly by the supraspinatus, inferiorly by the subscapularis, and with the coracoid process forming its base [8]. Key structures within this interval include the coracohumeral ligament (CHL), the superior glenohumeral ligament (SGHL), the long head of the biceps tendon, and the anterior joint capsule. These components play a critical role in maintaining the stability of the glenohumeral joint. Contracture of the rotator interval along with fibrosis, hyalinization, and fibroid degeneration of the CHL have been linked to the loss of shoulder external rotation in adhesive capsulitis [5,9]. On the other hand, studies have found elevated levels of inflammatory mediators not only in the joint capsule but also in the subacromial-subdeltoid bursa (SSB) [10,11,12,13].

There is a wide variety of treatments available for adhesive capsulitis, ranging from non-operative approaches such as nonsteroidal anti-inflammatory drugs (NSAIDs), physiotherapy, corticosteroid injections, and joint hydrodilatation, to surgical options like manipulation under anesthesia and arthroscopic capsular release [14]. Corticosteroids exert a general suppressive effect on the inflammatory response and inhibit the differentiation of fibroblasts into myofibroblasts. Given that both the SSB and the CHL play a role in the development of adhesive capsulitis—with the CHL primarily contributing to the loss of external rotation—we adopted an extra-articular approach. These two sites can be efficiently and accurately targeted under ultrasound guidance through the same puncture site.

The objective of this study was to assess the effectiveness of SSB and CHL corticosteroids injection in treating adhesive capsulitis, with a specific emphasis on examining the potential additional benefits of electrotherapy and conventional rehabilitation exercises.

## 2. Materials and Methods

### 2.1. Participants

Patients with unilateral shoulder pain and restricted ROM were recruited from outpatient clinics. Our study was approved by the institution review board of a tertiary medical center (IRB ID: 201407713A3C501). All patients signed informed consent before participation. Inclusion criteria were as follows: (1) aged between 20 and 70 years, (2) unilateral shoulder pain lasting more than one month with a Numerical Rating Scale (NRS) score greater than 3, and (3) restricted shoulder ROM in the coronal, sagittal, and frontal planes. The exclusion criteria were (1) sonography-confirmed supraspinatus tendon tear, (2) presence of calcified tendinopathy, (3) presence of cervical radiculopathy, (4) a history of tumors in the shoulder region, (5) receipt of corticosteroid injections or physical therapy for the affected shoulder within the previous three months, (6) a history of cerebrovascular accidents, and (7) allergy to corticosteroid injections.

A total of 29 patients, comprising 23 females and 6 males, were included in the study. The patients were subsequently divided into two groups. Group 1 consisted of 18 patients who received corticosteroid injections only, while Group 2 included 11 patients who received injections in combination with regular electrotherapy and conventional rehabilitation exercises during the 12-week follow-up period. Table 1 presents demographics and baseline characteristics for the two groups. These two groups were compatible with respect to age, height, weight, and body mass index (BMI), and no significant difference in shoulder ROM and SPADI at baseline was found in these two groups (*p* > 0.05).

### 2.2. Ultrasound Examination and Intervention

Shoulder ultrasound examination was conducted following standard protocol [15] using a Philips iU22 ultrasound machine equipped with a probe frequency range of 5–12 MHz. The same senior ultrasound operator performed all ultrasound examinations and injections in this study with dynamic focus, gain, and time-gain compensation settings fixed.

To perform ultrasound-guided corticosteroid injections targeting the SSB and CHL, patients were positioned in a side-lying posture, with the arm fully internally rotated and placed behind the back, and the elbow flexed. A round pillow was positioned under the upper back to ensure posture stability (Figure 1).

Two syringes of injectates were prepared, each containing a mixture of 7 mg betamethasone and 4 mL of 1% lidocaine. A 23-gauge, 2.5-inch needle was used for the injections. The ultrasound transducer was first placed over the lateral border of the coracoid process and supraspinatus tendon to obtain the axial oblique view (Figure 2a). The SSB was revealed between the deltoid muscle and supraspinatus tendon, and the CHL was seen as a hyperechoic band between the SSB and coracoid process (Figure 2b). The needle traveled through the SSB and ended up at the CHL to deliver the first injection (Figure 2c). The probe was then turned about 90 degrees and repositioned to the long-axis view of the supraspinatus tendon (Figure 3a). Afterward, the needle was redirected along the long-axis, from medial to lateral, targeting the SSB. The second injection was administered at this site (Figure 3b) [16]. This approach offered two key advantages: (1) The entire stretch of the CHL could be visualized, allowing precise drug delivery along the ligament; and (2) a single puncture site could be used to access both the CHL and SSB, minimizing patient discomfort during the procedure. If the patient could not achieve at least 80% of the ROM compared to the unaffected side in the sagittal, frontal, and transverse planes, an additional injection was administered after a four-week interval. The maximum number of injection sessions allowed was three. Acetaminophen was given in case of pain.

### 2.3. Clinical Assessment

The Shoulder Pain and Disability Index (SPADI) and shoulder ROM at the point where pain occurred were used as outcome measures. The clinical assessments were performed before the injection, 4 weeks (first endpoint), 8 weeks (second endpoint), and 12 weeks (third endpoint) following the initial injection of the SSB and CHL.

### 2.4. Shoulder Pain and Disability Index (SPADI)

SPADI is a self-administered questionnaire which evaluates pain and functional activities. It contains 13 questions: five addressing the severity of pain and eight focusing on the difficulty of daily living activities that involve upper-extremity use. [17,18]. The SPADI demonstrated satisfactory reliability and validity properties in patients with frozen shoulder [19]. The original version of the SPADI was well adapted and translated into Chinese. The Cronbach alpha ranged from 0.812 to 0.912 in all subscales and total scale of the Chinese-SPADI, indicating good or excellent internal consistency. The test–retest reliability (ICC = 0.887–0.915, SEM = 5.47, MDC = 15.16) was proven to be good or excellent [20]. Pain score, disability score, and the total SPADI score were calculated based on the patients’ responses.

### 2.5. Range of Motion (ROM) of Shoulder

Standard universal goniometric assessments were performed to evaluate bilateral shoulder ROM. All measurements were conducted by the same occupation therapist.

For shoulder flexion, patients were positioned supine. Goniometric measurements were taken with the fulcrum placed anterior to the acromion process, the stationary arm aligned parallel to the midaxillary line, and the moving arm aligned with the midline of the humerus. For shoulder extension, patients were positioned prone. The fulcrum was placed anterior to the acromion process, the stationary arm was aligned parallel to the midaxillary line, and the moving arm was aligned with the midline of the humerus. For shoulder internal rotation, patients were positioned supine with the shoulder abducted to 90 degrees and the elbow flexed to 90 degrees. The fulcrum was placed over the olecranon process of the ulna, with the stationary arm aligned vertically and the moving arm aligned with the ulnar styloid process. For shoulder external rotation, patients were also positioned supine with the shoulder abducted to 90 degrees and the elbow flexed to 90 degrees. The fulcrum was placed over the olecranon process of the ulna, the stationary arm aligned vertically, and the moving arm aligned with the ulnar styloid process. For shoulder abduction, patients were kept in prone position, and goniometric assessments were conducted by placing the fulcrum and body anterior to the acromion process, the stationary arm parallel to the midline of the trunk, and the moving arm aligned with the midline of the humeral bone. For shoulder abduction, patients were positioned prone. Goniometric assessments were performed by placing the fulcrum anterior to the acromion process, aligning the stationary arm parallel to the midline of the trunk, and aligning the moving arm with the midline of the humerus.

Riddle et al. revealed that intraclass correlation coefficients (ICCs) for intratester reliability were 0.98 for flexion, 0.98 for abduction, 0.94 for extension, 0.90 for horizontal adduction, 0.99 for lateral rotation, and 0.94 for medial rotation. ICCs reflecting intertester reliability were notably lower, ranging from 0.26 to 0.90 [21].

### 2.6. Statistical Analysis

Two sample *t*-test was used to compare differences of age, height, weight, BMI, and baseline ROM and SPADI between Group 1 and Group 2. Fisher’s exact test was used to compare the difference of gender distributions between the 2 groups.

Comparisons of ROM and SAPDI across over all time points between the 2 groups were assessed by mixed effects model with repeated measurements (MMRM) analysis. The MMRM analysis was based on unstructured variance–covariance matrix and denominator degree of freedom using the Kenward Roger method including a between-group fixed effect (Group 1 vs. Group 2) and a within-patient repeated factor (evaluation time: pre-treatment, 4 weeks after treatment, 8 weeks after treatment, and 12 weeks after treatment), as well as group-by-time interaction with individual patient as a cluster unit. Additionally, the MMRM model also included the ROM assessment from the untreated (normal) arm and time-by-ROM assessment from the untreated (normal) arm as covariates.

All significance levels were set at two-sided *p* < 0.05, and SAS Studio version 3.1M1 was used to perform all statistical analyses.

This is a preliminary proof-of-concept study; as such, the sample size is not powered. Further enrollment will be continued and the study will be powered for confirmatory purpose.

## 3. Results

### 3.1. Shoulder Pain and Disability Index (SPADI)

Table 2 and Figure 4 summarize the effect of time and group on pain score, disability score, and total SPADI score. Significant improvement over time was found for all of the three scores. However, the within-group comparison for these three scores illustrated that significant improvements were identified at all post-treatment timepoints in Group 1 but only at week 8 in Group 2. No significant time-by-group interaction was identified for pain scores. Group 1 tended to achieve more favorable outcomes on the disability score (*p* = 0.0723 for time-by-group interaction) and total score (*p* = 0.0877 for time-by-group interaction).

### 3.2. Shoulder Range of Motion (ROM)

Table 3 and Figure 5 summarize the active range of motion (AROM) of the shoulder. Significant improvements in flexion, abduction, and external rotation were observed in both groups at 4, 8, and 12 weeks post-treatment. However, no significant time-by-group interactions were detected for these movements. For internal rotation, Group 1 showed significant improvements at 4 and 12 weeks post-treatment, whereas Group 2 demonstrated significant improvements at all post-treatment time points. Despite these differences, the time-by-group interaction was not significant for internal rotation. Conversely, a significant time-by-group interaction was observed for extension (*p* = 0.0371). Group 1 showed no significant changes in extension across all time points, while Group 2 exhibited an increase in extension ROM at week 8 followed by a decrease at week 12. However, these changes in Group 2 were not statistically significant.

Table 4 and Figure 6 present the passive range of motion (PROM) of the shoulder. Significant improvements in flexion and abduction were observed in both groups. However, Group 1 showed significant improvements at all post-treatment time points, while Group 2 displayed improvements only at weeks 4 and 8. For internal rotation, significant improvements were observed across all post-treatment time points in both groups, with no significant time-by-group interaction detected. For external rotation, Group 1 demonstrated significant improvements across all post-treatment time points, while no significant improvements were noted in Group 2. The time-by-group interaction for external rotation was not significant (*p* = 0.4658). Regarding extension, no significant improvements were observed in Group 1 across all time points. In Group 2, extension ROM did not significantly change at weeks 4 and 8 but showed significant deterioration at week 12. A significant time-by-group interaction was noted for extension ROM (*p* = 0.0512).

Overall, these results illustrate that injection of corticosteroid provided significant improvements in ROM in both groups. Notably, eight patients (five from Group 1 and three from Group 2) achieved more than 80% ROM in all planes after the first injection and therefore did not require a second one. Following the second injection, 10 patients (five from each group) reached more than 80% ROM compared to the unaffected side in all planes and subsequently did not require further treatment.

## 4. Discussion

The main finding of this study was that physiotherapy did not enhance the effectiveness of a single percutaneous ultrasound-guided injection to the SSB and CHL in treating adhesive capsulitis. This injection approach resulted in significant improvements in pain, upper extremity function, and ROM across the frontal, sagittal, and coronal planes.

Pain reduction as well as functional improvement and restoration in flexion, abduction, and external rotation ROM were observed after a single ultrasound-guided steroid injection to the SSB and CHL. Additional improvement of internal rotation ROM was seen with repeat injections in both Group 1 and Group 2. Therefore, a thorough evaluation of the ROM limitation should be performed prior to the injection, and the consecutive injections might be needed when the limitation is severe and internal rotation limitation is considerably serious.

Dual-target ultrasound corticosteroid injections have been purposely repeatedly in order to enhance treatment effect and lower patient discomfort. This approach most commonly targets the SSB/supraspinatus tendon and the long head of the biceps tendon. It has shown to extend the duration of pain relief in shoulder impingement cases and has produced greater improvements in pain, ROM, and function in hemiplegic shoulders [22,23]. The distinct pathology of adhesive capsulitis warrants a different dual-target approach, focusing on the SSB and CHL, which can be achieved through a single injection following our protocol. With thorough anatomical knowledge and the aid of ultrasound guidance, direct puncture of the tendon can be avoided. This allows corticosteroids to suppress inflammation and facilitate tissue remodeling to their fullest potential.

The optimal injection strategy for adhesive capsulitis remains a subject of debate. Hydrodilatation of the glenohumeral joint is a long-established treatment for adhesive capsulitis, aimed at stretching the contracted joint cavity and reducing intra-articular inflammation. A Cochrane systematic review analyzing five studies found that only one had a low risk of bias. This study demonstrated that joint distension with saline and corticosteroids was more effective than a placebo in improving pain, function, and ROM at the three-week follow-up. However, the benefits were inconsistent at 6 and 12 weeks [24]. In a 2019 meta-analysis, Shang et al. [25] compared the effects of intra-articular and subacromial corticosteroid injections in patients with adhesive capsulitis. The study found no significant differences in flexion, abduction, and external ROM up to 24 weeks, nor in VAS pain scores beyond the three-week mark. Additionally, a recent prospective clinical trial involved 54 patients with adhesive capsulitis, who were administered 1 mL of triamcinolone acetonide (20 mg/mL) and 1 mL of prilocaine (2%) either anteriorly, between the CHL and the biceps tendon, or posteriorly into the joint [26]. Improvements in pain, function, and active ROM were similar between the two groups during the six-week follow-up period. Our findings aligned with the existing literature and underscored the effectiveness of extra-articular injections in the management of adhesive capsulitis. Furthermore, intra-articular communication with the subacromial space may occur in cases of adhesive capsulitis, potentially leading to corticosteroid leakage and providing additional therapeutic effects. Clinicians should consider both intra-articular and extra-articular injection techniques as viable treatment options for managing patients with this condition.

Interestingly, in the group combining corticosteroids injection and rehabilitation, the restoration of flexion was impeded and a second injection was necessitated. Meanwhile, abduction ROM failed to increase at the second and third endpoints. A meta-analysis based on two studies involving 86 participants suggested that adding manual therapy, exercise, and electrotherapy to intra-articular corticosteroid injections did not result in better outcomes in terms of pain, function, or quality of life [27]. Moreover, according to a meta-analysis including 2402 participants, intra-articular corticosteroids injection was only superior to physiotherapy for ER ROM during the first 6 weeks [28]. The results for pain, function, and mid- to long-term ROM were similar between the two approaches [28]. Moreover, cost-effectiveness analysis indicated that corticosteroid injections alone may be more cost-effective than corticosteroids with adjuvant physiotherapy or physiotherapy alone [29]. It is important to note that the nature and duration of rehabilitation programs vary greatly, and manipulation techniques are dependent on the therapist’s expertise. The long-term outcomes of physiotherapy remain uncertain due to the limited evidence. Currently, there are no definitive guidelines for the clinical management of adhesive capsulitis. Nevertheless, our results emphasize the promise and cost-effectiveness of using extra-articular corticosteroid injections as a standalone treatment option.

There were several limitations in this study. First, the sample size was relatively small, which may limit the generalizability of the findings. Second, the follow-up period was restricted to only three months, which may not have been sufficient to fully capture the long-term effects of the treatment or potential relapses. Third, we did not conduct subgroup analyses based on the stage of the disease (e.g., freezing, frozen, and thawing stages), which may have influenced the treatment outcomes. Future research should include patients at various disease stages and follow them for longer periods to allow for a more comprehensive assessment. Finally, conclusions regarding the optimal type or duration of rehabilitative intervention could not be drawn from this study. It remains unclear whether certain rehabilitation protocols are more effective than others when combined with injection therapy. Further studies are necessary to identify the most effective rehabilitation strategies to maximize patient recovery.

## 5. Conclusions

Physiotherapy did not provide additional benefits for patients with adhesive capsulitis beyond those achieved with a dual-target, ultrasound-guided corticosteroid injection targeting the SSB and CHL. The injection alone was effective in decreasing shoulder pain and disability. Significant improvements in shoulder ROM were also achieved. This approach may serve as a viable treatment option for affected patients. Because of the small sample size of this study, larger-scale research to optimize current physiotherapy with electrotherapy and traditional rehabilitation exercises after ultrasound-guided corticosteroid injections is warranted.

## Figures and Tables

**Figure 1 biomedicines-12-02668-f001:**
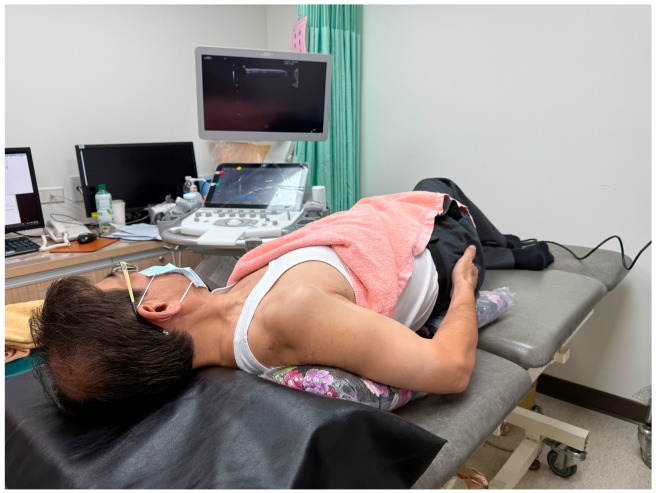
Patient positioning.

**Figure 2 biomedicines-12-02668-f002:**
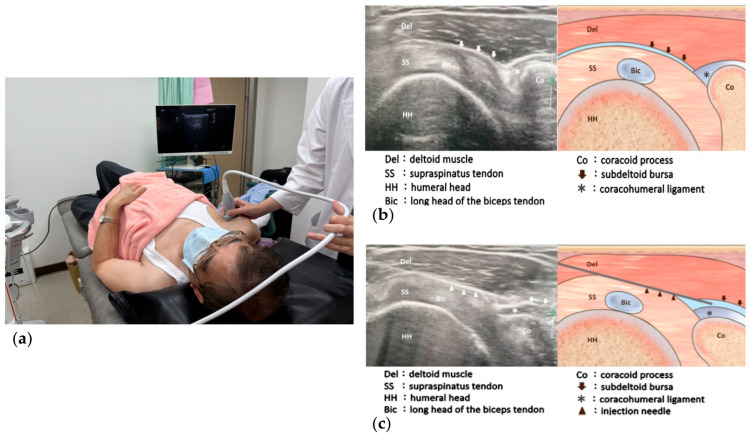
(**a**) Patient positioning and transducer placement for ultrasound-guided injection to the coracohumeral ligament (CHL); (**b**) Ultrasound image and illustration of the supraspinatus tendon axial oblique view; (**c**) Ultrasound image and illustration of the needle targeting the subacromial-subdeltoid bursa and CHL.

**Figure 3 biomedicines-12-02668-f003:**
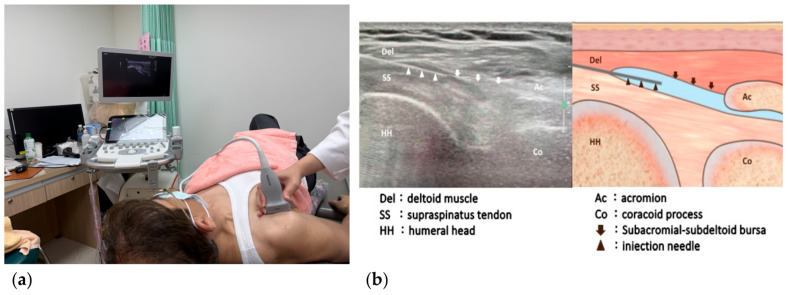
(**a**) Patient positioning and transducer placement for ultrasound-guided injection to the subacromial-subdeltoid bursa (SSB); (**b**) Ultrasound image and illustration of the needle passing through the SSB along the long axis of supraspinatus tendon.

**Figure 4 biomedicines-12-02668-f004:**
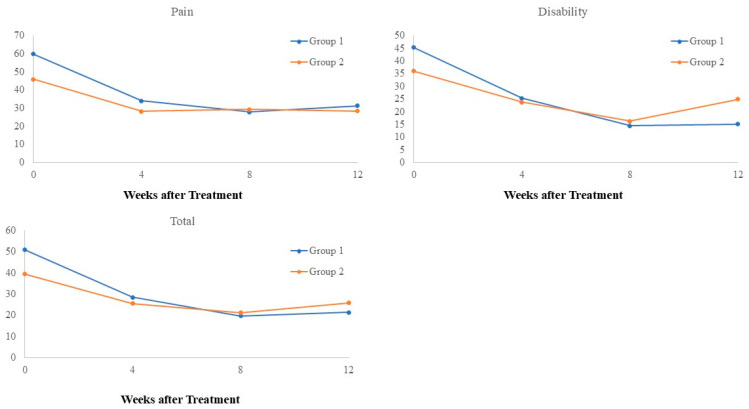
Effect of time and group on SPADI.

**Figure 5 biomedicines-12-02668-f005:**
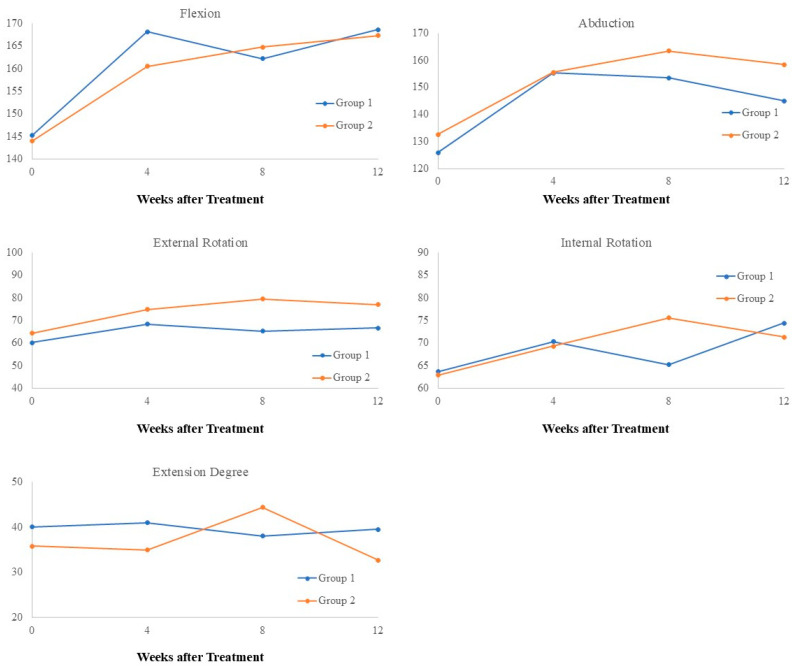
Effect of time and group on AROM.

**Figure 6 biomedicines-12-02668-f006:**
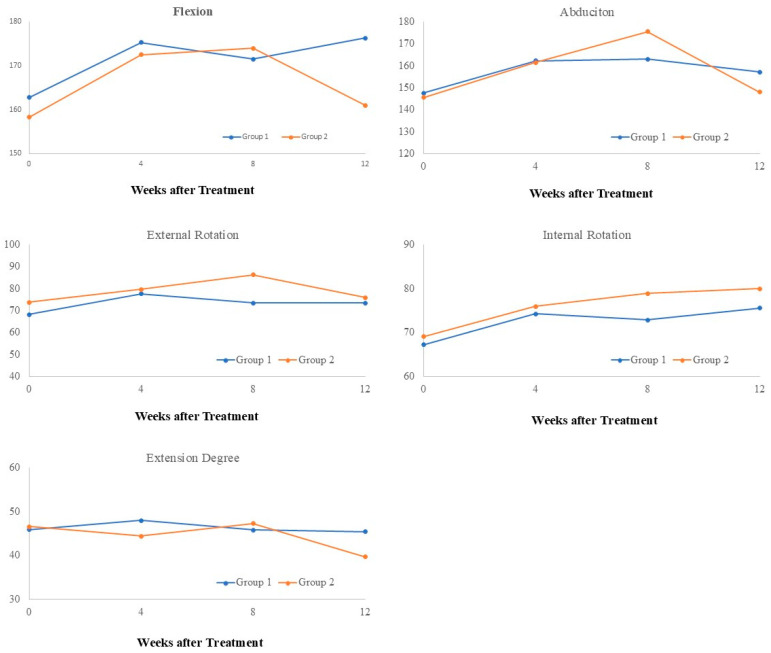
Effect of time and group on PROM.

**Table 1 biomedicines-12-02668-t001:** Demographic data.

Demographics and Baseline Characteristics	Group 1 (N = 18)	Group 2 (N = 11)	*p*-Value
Age (year), Mean ± SD	56.9 ± 7.62	53.0 ± 12.70	0.3098
Gender			
Men, n (%)	4 (22.2%)	2 (18.2%)	1.0000
Women, n (%)	14 (77.8%)	9 (81.8%)	
Weight (kg), Mean ± SD	57.1 ± 12.70	57.9 ± 8.17	0.8591
Height (cm), Mean ± SD	157.5 ± 7.66	158.3 ± 6.78	0.7855
BMI (kg/m^2^), Mean ± SD	22.9 ± 3.65	23.0 ± 1.96	0.8600
AROM			
Flexion, Mean ± SD	145.2 ± 36.84	144.0 ± 23.21	0.9224
Abduction, Mean ± SD	125.9 ± 40.68	132.6 ± 44.10	0.6778
External rotation, Mean ± SD	60.2 ± 25.11	64.3 ± 22.40	0.6603
Internal rotation, Mean ± SD	63.7 ± 16.89	62.9 ± 18.02	0.9033
Extension, Mean ± SD	40.1 ± 7.80	35.8 ± 11.81	0.2532
PROM			
Flexion, Mean ± SD	162.8 ± 21.23	158.3± 23.89	0.6011
Abduction, Mean ± SD	147.5 ± 39.53	145.5 ± 40.52	0.8944
External rotation, Mean ± SD	68.2 ± 26.54	73.8 ± 24.14	0.5700
Internal rotation, Mean ± SD	67.2 ± 14.27	69.1 ± 16.12	0.7470
Extension, Mean ± SD	45.9 ± 6.38	46.6 ± 7.97	0.7983
SAPDI			
Pain, Mean ± SD	59.8 ± 21.80	45.9 ± 27.43	0.1430
Disability, Mean ± SD	45.4 ± 30.15	36.1 ± 21.64	0.3786
Total, Mean ± SD	51.0 ± 25.94	39.6 ± 20.77	0.2296

BMI, body mass index; AROM, active range of motion; PROM, passive range of motion; SPADI, shoulder pain and disability index; SD, standard deviation.

**Table 2 biomedicines-12-02668-t002:** Shoulder Pain and Disability Index.

Measurement	Group	Statistics	Pre-Treatment n Mean (SD)	4-Week n Mean (SD)	8-Week n Mean (SD)	12-Week n Mean (SD)	Time Effect *p*-Value	Group Effect *p*-Value	Time & Group Interaction *p*-Value
SPADI									
Pain	Group 1 (N = 18)	Summary Statistics	18 59.8 (21.80)	18 34.0 (23.09)	13 27.9 (14.31)	8 31.4 (25.33)	0.0004 *	0.7228	0.4720
		Time effect Within Group: *p*-value	—Reference—	0.0018 *	<0.0001 *	0.0014 *			
	Group 2 (N = 11)	Summary Statistics	11 45.9 (27.43)	11 28.2 (21.95)	8 29.4 (31.73)	3 28.4 (38.87)			
		Time effect Within Group: *p*-value	—Reference—	0.0761	0.0201 *	0.5207			
	Total (N = 29)	Summary Statistics	29 54.5 (24.58)	29 31.8 (22.45)	21 28.5 (21.81)	11 30.6 (27.45)			
		Time effect Within Group: *p*-value	—Reference—	0.0013 *	<0.0001 *	0.0133 *			
Disability	Group 1 (N = 18)	Summary Statistics	18 45.4 (30.15)	18 25.3 (21.83)	13 14.5 (14.87)	8 15.2 (20.87)	0.0015 *	0.8181	0.0723
		Time effect Within Group: *p*-value	—Reference—	0.0029 *	<0.0001 *	<0.0001 *			
	Group 2 (N = 11)	Summary Statistics	11 36.1 (21.64)	11 23.8 (21.65)	8 16.3 (20.05)	3 24.9 (29.13)			
		Time effect Within Group: *p*-value	—Reference—	0.1319	0.0310 *	0.2281			
	Total (N = 29)	Summary Statistics	29 41.9 (27.21)	29 24.8 (21.38)	21 15.2 (16.56)	11 17.8 (22.26)			
		Time effect Within Group: *p*-value	—Reference—	0.0031 *	<0.0001 *	0.0003 *			
Total	Group 1 (N = 18)	Summary Statistics	18 51.0 (25.94)	18 28.5 (20.14)	13 19.8 (14.06)	8 21.4 (22.09)	0.0004 *	0.9544	0.0877
		Time effect Within Group: *p*-value	—Reference—	0.0013 *	<0.0001 *	<0.0001 *			
	Group 2 (N = 11)	Summary Statistics	11 39.6 (20.77)	11 25.6 (19.32)	8 21.3 (23.50)	3 25.9 (32.16)			
		Time effect Within Group: *p*-value	—Reference—	0.0919	0.0271 *	0.3315			
	Total (N = 29)	Summary Statistics	29 46.7 (24.37)	29 27.4 (19.54)	21 20.3 (17.67)	11 22.6 (23.51)			
		Time effect Within Group: *p*-value	—Reference—	0.0013 *	<0.0001 *	0.0008 *			

N = Total number of patients enrolled into each group. n = Number of patients with non-missing data. SPADI, shoulder pain and disability index; SD, standard deviation. * Statistical significance (*p* < 0.05).

**Table 3 biomedicines-12-02668-t003:** Effect of time and group on AROM.

Measurement	Group	Statistics	Pre-Treatment n Mean (SD)	4-Week n Mean (SD)	8-Week n Mean (SD)	12-Week n Mean (SD)	Time Effect *p*-Value	Group Effect *p*-Value	Time & Group Interaction *p*-Value
AROM									
Flexion	Group 1 (N = 18)	Summary Statistics	18 145.2 (36.84)	18 168.2 (9.12)	13 162.2 (14.78)	8 168.6 (8.94)	0.0184 *	0.4000	0.5033
		Time effect Within Group: *p*-value	—Reference—	0.0073 *	0.0249 *	0.0084 *			
	Group 2 (N = 11)	Summary Statistics	11 144.0 (23.21)	11 160.5 (16.77)	8 164.8 (7.17)	3 167.3 (15.53)			
		Time effect Within Group: *p*-value	—Reference—	0.0756	0.0496 *	0.0202 *			
	Total (N = 29)	Summary Statistics	29 144.8 (31.89)	29 165.3 (12.85)	21 163.2 (12.27)	11 168.3 (10.23)			
		Time effect Within Group: *p*-value	—Reference—	0.0030 *	0.0046 *	0.0009 *			
Abduction	Group 1 (N = 18)	Summary Statistics	18 125.9 (40.68)	18 155.2 (32.85)	13 153.5 (38.43)	8 145.0 (36.61)	0.0470 *	0.8575	0.8980
		Time effect Within Group: *p*-value	—Reference—	0.0020 *	0.0037 *	0.0217 *			
	Group 2 (N = 11)	Summary Statistics	11 132.6 (44.10)	11 155.5 (33.59)	8 163.4 (16.77)	3 158.3 (13.50)			
		Time effect Within Group: *p*-value	—Reference—	0.0304 *	0.0651	0.0558			
	Total (N = 29)	Summary Statistics	29 128.4 (41.35)	29 155.3 (32.53)	21 157.2 (31.76)	11 148.6 (31.83)			
		Time effect Within Group: *p*-value	—Reference—	0.0006 *	0.0018 *	0.0065 *			
External rotation	Group 1 (N = 18)	Summary Statistics	18 60.2 (25.11)	18 68.4 (19.32)	13 65.3 (20.87)	8 66.6 (17.23)	0.0090 *	0.3508	0.9820
		Time effect Within Group: *p*-value	—Reference—	0.0093 *	0.0047 *	0.0190 *			
	Group 2 (N = 11)	Summary Statistics	11 64.3 (22.40)	11 74.9 (15.58)	8 79.5 (10.45)	3 77.0 (15.87)			
		Time effect Within Group: *p*-value	—Reference—	0.0199 *	0.0066 *	0.0389 *			
	Total (N = 29)	Summary Statistics	29 61.7 (23.79)	29 70.9 (18.00)	21 70.7 (18.70)	11 69.5(16.78)			
		Time effect Within Group: *p*-value	—Reference—	0.0010 *	0.0002 *	0.0034 *			
Internal rotation	Group 1 (N = 18)	Summary Statistics	18 63.7 (16.89)	18 70.3 (11.82)	13 65.2 (20.54)	8 74.4 (9.04)	0.0006 *	0.9095	0.1298
		Time effect Within Group: *p*-value	—Reference—	0.0533 *	0.2020	0.0003 *			
	Group 2 (N = 11)	Summary Statistics	11 62.9 (18.02)	11 69.3 (8.71)	8 75.6 (7.29)	3 71.3 (10.02)			
		Time effect Within Group: *p*-value	—Reference—	0.0520 *	0.0042 *	0.0035 *			
	Total (N = 29)	Summary Statistics	29 63.4 (17.01)	29 69.9 (10.59)	21 69.1 (17.29)	11 73.5 3(8.90)			
		Time effect Within Group: *p*-value	—Reference—	0.0086 *	0.0031 *	<0.0001 *			

N = Total number of patients enrolled into each group. n = Number of patients with non-missing data. AROM, active range of motion; SD, standard deviation. * Statistical significance (*p* < 0.05).

**Table 4 biomedicines-12-02668-t004:** Effect of time and group on PROM.

Measurement	Group	Statistics	Pre-Treatment n Mean (SD)	4-Week n Mean (SD)	8-Week n Mean (SD)	12-Week n Mean (SD)	Time Effect *p*-Value	Group Effect *p*-Value	Time & Group Interaction *p*-Value
PROM									
Flexion	Group 1 (N = 18)	Summary Statistics	18 162.8 (21.23)	18 175.3 (7.36)	13 171.5 (13.41)	8 176.3 (6.94)	<0.0001 *	0.1568	0.1219
		Time effect Within Group: *p*-value	—Reference—	0.0049 *	0.0113 *	0.0096 *			
	Group 2 (N = 11)	Summary Statistics	11 158.3 (23.89)	11 172.5 (13.24)	8 174.0 (6.91)	3 161.0 (10.54)			
		Time effect Within Group: *p*-value	—Reference—	0.0114 *	0.0080 *	0.3369 *			
	Total (N = 29)	Summary Statistics	29 161.1 (21.96)	29 174.3 (9.87)	21 172.5 (11.23)	11 172.1 (10.33)			
		Time effect Within Group: *p*-value	—Reference—	0.0004 *	0.0005 *	0.0199 *			
Abduction	Group 1 (N = 18)	Summary Statistics	18 147.5 (39.53)	18 162.2 (31.16)	13 163.1 (34.07)	8 157.1 (33.93)	0.0167 *	0.7307	0.6298
		Time effect Within Group: *p*-value	—Reference—	0.0682	0.0783	0.0538			
	Group 2 (N = 11)	Summary Statistics	11 145.5 (40.52)	11 161.5 (26.64)	8 175.6 (10.50)	3 148.0 (25.06)			
		Time effect Within Group: *p*-value	—Reference—	0.0701	0.0397 *	0.9031			
	Total (N = 29)	Summary Statistics	29 146.7 (39.20)	29 161.9 (29.04)	21 167.9 (27.82)	11 154.6 (30.82)			
		Time effect Within Group: *p*-value	—Reference—	0.0515 *	0.0497 *	0.2377			
External rotation	Group 1 (N = 18)	Summary Statistics	18 68.2 (26.54)	18 77.6 (20.05)	13 73.4 (18.86)	8 73.5 (15.84)	0.0708	0.7807	0.4658
		Time effect Within Group: *p*-value	—Reference—	0.0031 *	0.0177 *	0.0572 *			
	Group 2 (N = 11)	Summary Statistics	11 73.8 (24.14)	11 79.7 (15.23)	8 86.3 (6.94)	3 76.0 (7.03)			
		Time effect Within Group: *p*-value	—Reference—	0.1604	0.0778	0.7287			
	Total (N = 29)	Summary Statistics	29 70.3 (25.37)	29 78.4 (18.11)	21 78.3 (16.47)	11 74.2 (16.53)			
		Time effect Within Group: *p*-value	—Reference—	0.0038 *	0.0054 *	0.1682			
Internal rotation	Group 1 (N = 18)	Summary Statistics	18 67.2 (14.27)	18 74.2 (9.43)	13 71.9 (13.89)	8 75.6 (7.29)	0.0012 *	0.3953	0.7304
		Time effect Within Group: *p*-value	—Reference—	0.0334 *	0.0703	0.0610			
	Group 2 (N = 11)	Summary Statistics	11 69.1 (16.12)	11 76.0 (7.03)	8 78.9 (5.08)	3 80.0 (12.00)			
		Time effect Within Group: *p*-value	—Reference—	0.0450 *	0.0690	0.0245 *			
	Total (N = 29)	Summary Statistics	29 67.9 (14.74)	29 74.9 (8.51)	21 74.6 (11.69)	11 76.8 (8.38)			
		Time effect Within Group: *p*-value	—Reference—	0.0041 *	0.0114 *	0.0047 *			
Extension	Group 1 (N = 18)	Summary Statistics	18 45.9 (6.38)	18 48.0 (5.18)	13 45.8 (8.86)	8 45.4 (4.27)	0.0835	0.1168	0.0512
		Time effect Within Group: *p*-value	—Reference—	0.5562	0.4815	0.3571			
	Group 2 (N = 11)	Summary Statistics	11 46.6 (7.97)	11 44.4 (8.63)	8 47.3 (7.69)	3 39.7 (9.71)			
		Time effect Within Group: *p*-value	—Reference—	0.0957	0.3259	0.0182 *			
	Total (N = 29)	Summary Statistics	29 46.2 (6.89)	29 46.6 (6.79)	21 46.4 (8.26)	11 43.8 (6.33)			
		Time effect Within Group: *p*-value	—Reference—	0.3452	0.2302	0.0141 *			

N = Total number of patients enrolled into each group. n = Number of patients with non-missing data. PROM, passive range of motion; SD, standard deviation. * Statistical significance (*p* < 0.05).

## Data Availability

The original contributions presented in the study are included in the article. Further inquiries can be directed to the corresponding author.

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
