# Peer review of "Evaluating the Effectiveness of Ultrasound-Guided Subacromial-Subdeltoid Bursa and Coracohumeral Ligament Corticosteroid Injections With and Without Physiotherapy in Adhesive Capsulitis Treatment"

_biomedicines, 2024, doi:10.3390/biomedicines12122668_

Round 1

Reviewer 1 Report

Comments and Suggestions for Authors

Dear authors 

Congratulations on the presented manuscript. I find the following concerns in your submission that prevent me from considering it for publication in its current state. May be following addressing them appropriately might warrant publication

1. The novelty of the study is not explained well apart from in the title 

2. The study is not powered enough to claim strong conclusions as stated in the manuscript 

3. The positioning of the patient could be better demonstrated for the benefit of the readers and the current image is only confusing

4. The comparisons were made in both the groups to the baseline and the significant improvement of which is naturally explained but the comparison between the groups has not been done or given in the results as expressed in the methods 

5. Intragroup comparison statistical methods were not elaborated in the methods section 

Comments on the Quality of English Language

the statements were concluded abrupt rather than a narrative style in many instances in methods and discussion which could be rephrased 

Author Response

Comment 1: The novelty of the study is not explained well apart from in the title

Response 1:

Thank for your comment! We changed the title.

The Effect of Ultrasound-Guided Subacromial -Subdeltoid Bursa and Coracoid Humeral Ligament Corticosteroid Injection in Treating Adhesive Capsulitis and descripted in Ultrasound examination and intervention

This approach offered two features: (1) We can see the whole stretch of CHL and delivered the drug along the ligament precisely; (2) We can use only one puncture site to reach the CHL and SSB in both transverse and frontal planes to reduce pain of the patients during this procedure.

We explain more details in the article(ultrasound examination and intervention).

The probe was then turned about 90 degree and repositioning to the long-axis view of the supraspinatus tendon (Figure 3a). The needle was redirected along the long-axis, from medial to lateral, targeting the SSB. The second injection was administered at this site with another syringe which contained 7 mg betamethasone and 4 mL of 1% lidocaine (Figure 3b)16. This approach offered two features: First, we can see the whole stretch of CHL and delivered the drug along the ligament precisely; Second, we can use only one puncture site to reach the CHL and SSB in both transverse and frontal planes to reduce pain of the patients during the procedure.

We also descripted in the conclusion of the abstract

Ultrasound-guided corticosteroid injection to the CHL and SSB significantly improved pain, disability, and ROM in patients with adhesive capsulitis. This extra-articular, dual-target approach may be a viable alternative treatment option to consider. Further research is needed to determine the optimal type and duration of rehabilitation programs for managing this condition.

Comment 2: The study is not powered enough to claim strong conclusions as stated in the manuscript

Response 2:

Thank for your comments!

This is a preliminary proof of concept study as such the sample size is not powered. Further enrollment will be continued and the study will be powered for confirmatory purpose.

Comment 3: The positioning of the patient could be better demonstrated for the benefit of the readers and the current image is only confusing

Response 3:

Thank for your comments! We add Figure 1 for better demonstration of positing of the patients.

To perform ultrasound-guided corticosteroid injection of the SSB and CHL, patients were positioned in a side-lying posture, with the arm fully internally rotated and placed behind the back, and the elbow flexed. A round pillow was positioned under the upper back to ensure posture stability (Figure 1).

Comment 4: The comparisons were made in both the groups to the baseline and the significant improvement of which is naturally explained but the comparison between the groups has not been done or given in the results as expressed in the methods

Response 4:

Thank for your comment! Statistical method had been written once again.

The assessment for between-group comparison based on mixed effect model with repeated measurement (MMRM) has been added under the statistical analysis section and the results for between-group comparisons have been added correspondingly for each endpoint.

Table 1. Demographic data.

Demographics and

Baseline Characteristics

Group 1

(N = 18)

Group 2

(N = 11)

P-value a

Age (year), Mean ± SD

56.9 ± 7.62

53.0 ± 12.70

0.3098

Gender

   Men, n (%)

4 (22.2%)

2 (18.2%)

1.0000

   Women, n (%)

14 (77.8%)

9 (81.8%)

Weight (kg), Mean ± SD

57.1 ± 12.70

57.9 ± 8.17

0.8591

Height (cm), Mean ± SD

157.5 ± 7.66

158.3 ± 6.78

0.7855

BMI (kg/m2), Mean ± SD

22.9 ± 3.65

23.0 ± 1.96

0.8600

AROM

   Flexion, Mean ± SD

145.2 ± 36.84

144.0 ± 23.21

0.9224

   Abduction, Mean ± SD

125.9 ± 40.68

132.6 ± 44.10

0.6778

   External rotation, Mean ± SD

60.2 ± 25.11

64.3 ± 22.40

0.6603

   Internal rotation, Mean ± SD

63.7 ± 16.89

62.9 ± 18.02

0.9033

   Extension, Mean ± SD

40.1 ± 7.80

35.8 ± 11.81

0.2532

PROM

   Flexion, Mean ± SD

162.8 ± 21.23

158.3± 23.89

0.6011

   Abduction, Mean ± SD

147.5 ± 39.53

145.5 ± 40.52

0.8944

   External rotation, Mean ± SD

68.2 ± 26.54

73.8 ± 24.14

0.5700

   Internal rotation, Mean ± SD

67.2 ± 14.27

69.1 ± 16.12

0.7470

   Extension, Mean ± SD

45.9 ± 6.38

46.6 ± 7.97

0.7983

SAPDI

   Pain, Mean ± SD

59.8 ± 21.80

45.9 ± 27.43

0.1430

   Disability, Mean ± SD

45.4 ± 30.15

36.1 ± 21.64

0.3786

   Total, Mean ± SD

51.0  ± 25.94

39.6  ± 20.77

0.2296

Values expressed as mean± standard devia*tion (SD) or numbers (%).

Abbreviations: BMI, body mass index; AROM, active range of motion; PROM, passive range of motion;

SPADI, shoulder pain and disability index.

*Statistical significance level set at p< .05 (2-sided).

a P-value based on Fisher’s exact test for gender, and two sample t-test for the remaining variables including age, weight, height, BMI, AROM, PROM, and SPADI.2.2.

Comment 5: Intragroup comparison statistical methods were not elaborated in the methods section

Response 5:

Thank for your comment! Statistical method had been written once again.

Similar to item 4) above, the method for intragroup comparison based MMRM has been added under the statistical analysis section and the results for within-group comparisons over time have been added correspondingly for each endpoint.

Comment 6: The statements were concluded abrupt rather than a narrative style in many instances in methods and discussion which could be rephrased.

Response 6: Thank for your comment! We made amendment for the whole article.

Conclusion: A dual-target, ultrasound-guided corticosteroids injection to the SSB and CHL was effective in decreasing shoulder pain and disability in patients with adhesive capsulitis. Significant improvements in shoulder ROM were also achieved. Therefore, this approach may serve as an alternative treatment option for affected patients.

Reviewer 2 Report

Comments and Suggestions for Authors

the authors report the treatment of a small group of patients using a series of Subdeltoid Bursa and Coracoid Humeral Ligament  Corticosteroid Injections.

it is preferable to use the term adhesive capsulitis and not the term frozen shoulder.

The main criticism is that the groups are not comparable and the follow up short.

The author approach is not novel since injecting both sites is common clinical practice.

As shown in Figure 1 you performed and extra and not an intra-articular steroid injection with small volume. Please be more precise and justify your practice.

15 effect of novel….a novel

 20 following standard protocol….a

 shoulder when pain occurred…Please be more specific.

And no additional benefits of electrotherapy and traditional rehabilitation exercise was observed. The conclusion is not absolutely valid given the limitations of your study.

Remove numbering of the keywords

 cost huge burden. Please restructure the whole sentence

Line 37. Reference #1 is not sufficient. Please add references mor

 of physiatrists and cost huge burden to.. Please restructure the whole sentence

39 first described 

The coracohumeral ligament can be evaluated with ultrasound but not the joint capsule and the axillary fold. Please be more precise

 researches of magnetic resonance imaging . Please restructure the whole sentence

exercise program has long been..

references [14,15] refer to subacromial impingement syndrome and not to frozen shoulder and are therefore not applicable. Please use other references.

references [16] is not relevant. Please correct.

Line 55. Incomprehensible. Please improve

observed between…add the two

As previous discussion…

Change..plural

Line 90. If you exclude patients with diabetes you instantly exclude 70 percent of the patients with primary adhesive capsulitis.

composed of.. included

lines 98-100. The range of motion in patients with adhesive capsulitis is very important. Please include ROM data in your materials.

Line 105. What is the probe frequency you are using?

Line 107. Please be more specific.

Line 112. The total injectate volume of 5ml is not large. Do you think that use of a larger volume could be beneficial?

You performed two different injections after repositioning of the needle. How much of the injectate volume did you inject in each site? Do you think that only 5 ml is sufficient for a double site injection?

In figure 1a the ultrasound probe is positioned directly over the biceps tendon and perpendicular to the humeral head. In figure 1b the biceps tendon is not shown. Please verify.

What was the post injection treatment regime? The injection group did no physiotherapy but received what kind of medication?

Is the SPADI questionnaire validated in the patient’s native language?

Did you perform an inter- and intraexaminer validation of the goniometric assessment of shoulder ROM?

According to your data the restriction of shoulder ROM was mild. Do you think that inclusion of chronic patients with frozen shoulder and greater stiffness could have affected your results?

Line 200. Reference #21 is not relevant.

Line 203. restoration of flexion was impeded. This is not correct since the disease is chronic and resistant to treatment.

several literatures. Please improve

lines 224-226. The references are not relevant.

Lines 228-261. The paragraph is not relevant to the study. It is preferable to describe the various mobilization protocols in shoulder stiffness.

Comments on the Quality of English Language

extensive editing is required

Author Response

Comment 1:

It is preferable to use the term adhesive capsulitis and not the term frozen shoulder.

Response 1: Thank for your comment! We used the term adhesive capsulitis in the article

Comment 2:

The main criticism is that the groups are not comparable and the follow up short.

Response 2: Thank for your comment! We provided a new table for comparison of two group.

Table 1 presents demographics and baseline characteristics for the 2 gorups. These two groups were compatible with respect to age, height, weight, body mass index (BMI) and no significant difference of shoulder ROM and SPADI at baseline in these two groups(P>0.05).

Table 1. Demographic data.

Demographics and

Baseline Characteristics

Group 1

(N = 18)

Group 2

(N = 11)

P-value a

Age (year), Mean ± SD

56.9 ± 7.62

53.0 ± 12.70

0.3098

Gender

   Men, n (%)

4 (22.2%)

2 (18.2%)

1.0000

   Women, n (%)

14 (77.8%)

9 (81.8%)

Weight (kg), Mean ± SD

57.1 ± 12.70

57.9 ± 8.17

0.8591

Height (cm), Mean ± SD

157.5 ± 7.66

158.3 ± 6.78

0.7855

BMI (kg/m2), Mean ± SD

22.9 ± 3.65

23.0 ± 1.96

0.8600

AROM

   Flexion, Mean ± SD

145.2 ± 36.84

144.0 ± 23.21

0.9224

   Abduction, Mean ± SD

125.9 ± 40.68

132.6 ± 44.10

0.6778

   External rotation, Mean ± SD

60.2 ± 25.11

64.3 ± 22.40

0.6603

   Internal rotation, Mean ± SD

63.7 ± 16.89

62.9 ± 18.02

0.9033

   Extension, Mean ± SD

40.1 ± 7.80

35.8 ± 11.81

0.2532

PROM

   Flexion, Mean ± SD

162.8 ± 21.23

158.3± 23.89

0.6011

   Abduction, Mean ± SD

147.5 ± 39.53

145.5 ± 40.52

0.8944

   External rotation, Mean ± SD

68.2 ± 26.54

73.8 ± 24.14

0.5700

   Internal rotation, Mean ± SD

67.2 ± 14.27

69.1 ± 16.12

0.7470

   Extension, Mean ± SD

45.9 ± 6.38

46.6 ± 7.97

0.7983

SAPDI

   Pain, Mean ± SD

59.8 ± 21.80

45.9 ± 27.43

0.1430

   Disability, Mean ± SD

45.4 ± 30.15

36.1 ± 21.64

0.3786

   Total, Mean ± SD

51.0  ± 25.94

39.6  ± 20.77

0.2296

Values expressed as mean± standard devia*tion (SD) or numbers (%).

Abbreviations: BMI, body mass index; AROM, active range of motion; PROM, passive range of motion;

SPADI, shoulder pain and disability index.

*Statistical significance level set at p< .05 (2-sided).

a P-value based on Fisher’s exact test for gender, and two sample t-test for the remaining variables including age, weight, height, BMI, AROM, PROM, and SPADI.

Response2: The follow up was short.

We put it into limitation of the study.

Second, the follow-up period was restricted to only three months, which may not have been sufficient to fully capture the long-term effects of the treatment or potential relapses.

Comment 3:

The author approach is not novel since injecting both sites is common clinical practice.

Response 3: Thank for your comments! We changed the title.

The Effect of Ultrasound-Guided Subacromial -Subdeltoid Bursa and Coracoid Humeral Ligament Corticosteroid Injection in Treating Adhesive Capsulitis and descripted in Ultrasound examination and intervention

This approach offered two features: (1) We can see the whole stretch of CHL and delivered the drug along the ligament precisely; (2) We can use only one puncture site to reach the CHL and SSB in both transverse and frontal planes to reduce pain of the patients during this procedure.

Comment 4:

As shown in Figure 1 you performed and extra and not an intra-articular steroid injection with small volume. Please be more precise and justify your practice.

Response 4: Thank for your comment! We provided a new picture to demonstrate the position of the probe.

This extra-articular, dual-target approach may be a viable alternative treatment option to consider.

Comment 5:

15 effect of novel….a novel

20 following standard protocol….a

shoulder when pain occurred…Please be more specific.

Response 5: Thank for your comment! We corrected the sentence.

Given that both the SSB and the CHL play a role in the development of adhesive capsulitis—with the CHL primarily contributing to the loss of external rotation—we proposed an extra-articular approach. These two sites can be efficiently and accurately targeted under ultrasound guidance through the same puncture site.

Comment 6:

And no additional benefits of electrotherapy and traditional rehabilitation exercise was observed. The conclusion is not absolutely valid given the limitations of your study.

Response 6: Thank for your comment! We add limitation of the study and modified the conclusion.

There were several limitations in this study. First, the sample size was relatively small, which may limit the generalizability of the findings. Second, the follow-up period was restricted to only three months, which may not have been sufficient to fully capture the long-term effects of the treatment or potential relapses. Third, we did not conduct subgroup analyses based on the stage of the disease (e.g., freezing, frozen, and thawing stages), which may have influenced the treatment outcomes. Future research should include patients at various disease stages and follow them for longer periods to allow for a more comprehensive assessment. Finally, conclusions regarding the optimal type or duration of rehabilitative intervention could not be drawn from this study. It remains unclear whether certain rehabilitation protocols are more effective than others when combined with injection therapy. Further studies are necessary to identify the most effective rehabilitation strategies to maximize patient recovery.

Conclusions

A dual-target, ultrasound-guided corticosteroids injection to the SSB and CHL was effective in decreasing shoulder pain and disability in patients with adhesive capsulitis. Significant improvements in shoulder ROM were also achieved. Therefore, this approach may serve as an alternative treatment option for affected patients.

Comment 7:

Remove numbering of the keywords.

Response 7: Thank for your comment! We deleted the number.

Keywords: Ultrasound-guided injection, subacromial subdeltoid bursa, coracohumeral ligament, frozen shoulder, shoulder pain

Comment 8:

cost huge burden. Please restructure the whole sentence

Response 8: Thank for your comment! We restructured the sentence.

Moreover, cost-effectiveness analysis indicated that corticosteroid injections alone may be more cost-effective than corticosteroids with adjuvant physiotherapy or physiotherapy alone 28.

Comment 9:

Line 37. Reference #1 is not sufficient. Please add references more.

Response 9: Thank for your comment! We expanded this paragraph and add some more references.

Shoulder pain is the third most frequently encontered musculoskeletal complaint in clinics and imposes a significant burden on patients' daily activities as well as healthcare costs 1. Adhesive capsulitis is acommon shoulder disorder with a lifetime prevalence of 2–5%, particularly affecting middle-aged women 2, 3. Commonly referred to as "frozen shoulder," the condition is characterized by pain and progressive, global glenohumeral joint range of motion (ROM) limitation 3. Although adhesive capsulitis is generally considered a self-limiting disease that resolves within two to three years, symptoms can be debilitating and may persist in up to 50% of patient 4.

Comment 10:

of physiatrists and cost huge burden to.. Please restructure the whole sentence.

Response 10: Thank for your comment! We restructured the sentence.

Moreover, cost-effectiveness analysis indicated that corticosteroid injections alone may be more cost-effective than corticosteroids with adjuvant physiotherapy or physiotherapy alone 28. It is important to note that the nature and duration of rehabilitation programs vary greatly, and manipulation techniques are dependent on the therapist's expertise. The long-term outcomes of physiotherapy remain uncertain due to the limited evidence. Currently, there are no definitive guidelines for the clinical management of adhesive capsulitis. Nevertheless, our results highlighted the promise of using extra-articular corticosteroid injections as an independent treatment.

Comment 11:

39 first described.

The coracohumeral ligament can be evaluated with ultrasound but not the joint capsule and the axillary fold. Please be more precise

Response 11: Thank for your comment! We made amendment.

The rotator interval is a triangular anatomical space located at the anterosuperior aspect of the shoulder, bordered superiorly by the supraspinatus, inferiorly by the subscapularis, and with the coracoid process forming its base 8. Key structures within this interval include the coracohumeral ligament (CHL), the superior glenohumeral ligament (SGHL), the long head of the biceps tendon, and the anterior joint capsule. These components play a critical role in maintaining the stability of the glenohumeral joint. Contracture of the rotator interval along with fibrosis, hyalinization, and fibroid degeneration of the CHL has been linked to the loss of shoulder external rotation in adhesive capsulitis 5, 9.

Comment 12: researches of magnetic resonance imaging . Please restructure the whole sentence

exercise program has long been..

Response 12: Thank for your comment! We restructured the sentence.

There is a wide variety of treatments is available for adhesive capsulitis, ranging from non-operative approaches such as nonsteroidal anti-inflammatory drugs (NSAIDs), physiotherapy, corticosteroid injections, and joint hydrodilatation, to surgical options like manipulation under anesthesia and arthroscopic capsular release 14.  Corticosteroids exert a general suppressive effect on the inflammatory response and inhibit the differentiation of fibroblasts into myofibroblasts. Given that both the SSB and the CHL play a role in the development of adhesive capsulitis—with the CHL primarily contributing to the loss of external rotation—we proposed an extra-articular approach. These two sites can be efficiently and accurately targeted under ultrasound guidance through the same puncture site.

Comment 13:

references [14,15] refer to subacromial impingement syndrome and not to frozen shoulder and are therefore not applicable. Please use other references.

Response 13: Thank for your comment! We removed these two references.

Comment 14:

references [16] is not relevant. Please correct.

Line 55. Incomprehensible. Please improve

observed between…add the two

As previous discussion…

Change..plural

Line 90. If you exclude patients with diabetes you instantly exclude 70 percent of the patients with primary adhesive capsulitis.

Response 14: Thank for your comment! We removed reference [16]. And we made amendment.

Yes, it is difficult to recruited patients when we exclude patients with diabetes.

Comment 15:

composed of.. included

lines 98-100. The range of motion in patients with adhesive capsulitis is very important. Please include ROM data in your materials.

Response 15: Thank for your comment! We add a new table for ROM and SPADI.

Table 1. Demographic data.

Demographics and

Baseline Characteristics

Group 1

(N = 18)

Group 2

(N = 11)

P-value a

Age (year), Mean ± SD

56.9 ± 7.62

53.0 ± 12.70

0.3098

Gender

   Men, n (%)

4 (22.2%)

2 (18.2%)

1.0000

   Women, n (%)

14 (77.8%)

9 (81.8%)

Weight (kg), Mean ± SD

57.1 ± 12.70

57.9 ± 8.17

0.8591

Height (cm), Mean ± SD

157.5 ± 7.66

158.3 ± 6.78

0.7855

BMI (kg/m2), Mean ± SD

22.9 ± 3.65

23.0 ± 1.96

0.8600

AROM

   Flexion, Mean ± SD

145.2 ± 36.84

144.0 ± 23.21

0.9224

   Abduction, Mean ± SD

125.9 ± 40.68

132.6 ± 44.10

0.6778

   External rotation, Mean ± SD

60.2 ± 25.11

64.3 ± 22.40

0.6603

   Internal rotation, Mean ± SD

63.7 ± 16.89

62.9 ± 18.02

0.9033

   Extension, Mean ± SD

40.1 ± 7.80

35.8 ± 11.81

0.2532

PROM

   Flexion, Mean ± SD

162.8 ± 21.23

158.3± 23.89

0.6011

   Abduction, Mean ± SD

147.5 ± 39.53

145.5 ± 40.52

0.8944

   External rotation, Mean ± SD

68.2 ± 26.54

73.8 ± 24.14

0.5700

   Internal rotation, Mean ± SD

67.2 ± 14.27

69.1 ± 16.12

0.7470

   Extension, Mean ± SD

45.9 ± 6.38

46.6 ± 7.97

0.7983

SAPDI

   Pain, Mean ± SD

59.8 ± 21.80

45.9 ± 27.43

0.1430

   Disability, Mean ± SD

45.4 ± 30.15

36.1 ± 21.64

0.3786

   Total, Mean ± SD

51.0  ± 25.94

39.6  ± 20.77

0.2296

Values expressed as mean± standard devia*tion (SD) or numbers (%).

Abbreviations: BMI, body mass index; AROM, active range of motion; PROM, passive range of motion;

SPADI, shoulder pain and disability index.

*Statistical significance level set at p< .05 (2-sided).

a P-value based on Fisher’s exact test for gender, and two sample t-test for the remaining variables including age, weight, height, BMI, AROM, PROM, and SPADI.2.2.

Comment 16:

Line 105. What is the probe frequency you are using?

Response 16: Thank for your comments!

The frequency of the probe is 5-12 MHz. And we made amendment.

Shoulder ultrasound examination was conducted following standard protocol 15 using a Philips iU22 ultrasound machine equipped with a probe frequency range of 5–12 MHz.

Line 107. Please be more specific.

Line 112. The total injectate volume of 5ml is not large. Do you think that use of a larger volume could be beneficial?

You performed two different injections after repositioning of the needle. How much of the injectate volume did you inject in each site? Do you think that only 5 ml is sufficient for a double site injection?

Two syringes of injectates were prepared, each containing a mixture of 7 mg betamethasone and 4 mL of 1% lidocaine. A 23-gauge, 2.5-inch needle was used for the injection.

The needle travelled through SSB and ended up at CHL to deliver the first injection with a mixture of 7 mg betamethasone and 4 mL of 1% lidocaine. (Figure 2c). The probe was then repositioning to the long-axis view of the supraspinatus tendon (Figure 3a) and the needle was redirected along the long-axis, from medial to lateral, targeting the SSB. The second injection was administered at this site with another syringe which contain 7 mg betamethasone and 4 mL of 1% lidocaine. (Figure 3b)16.

Comment 17:

In figure 1a the ultrasound probe is positioned directly over the biceps tendon and perpendicular to the humeral head. In figure 1b the biceps tendon is not shown. Please verify.

Response 17: Thank for your comments! We provided a new photo for figure 1 to demonstrate positioning of the patient. On the other hand, we provided a new photo

to demonstrate the patient positioning and transducer placement for ultrasound-guided injection to the coracohumeral ligament (CHL) in figure 2(a). Supraspinatus tendon, biceps tendon and CHL can be seen this in axial oblique view in figure 2(b).

Figure 2. (a) Patient positioning and transducer placement for ultrasound-guided injection to the coracohumeral ligament (CHL); (b) Ultrasound image and illustration of the supraspinatus tendon axial oblique view; (c) Ultrasound image and illustration of the needle passing through the subacromial-subdeltoid bursa below the deltoid muscle and targeting the CHL.

Comment 18:

What was the post injection treatment regime? The injection group did no physiotherapy but received what kind of medication?

Response 18: Thank for your comment!

Acetaminophen was given for pain control.

Comment 19:

Is the SPADI questionnaire validated in the patient’s native language?

Response 19: Thank for your comment! These were study regarding validation in Chinese, and we put it into the article.

The SPADI demonstrated satisfactory reliability and validity properties in patients with frozen shoulder 19. The original version of the SPADI was well adapted and translated into Chinese. The Cronbach alpha ranged from 0.812 to 0.912 in all subscales and and total scale of the Chinese-SPADI, indicating good or excellent internal consistency. The test-retest reliability(ICC=0.887-0.915, SEM=5.47, MDC=15.16) was proved to be good or excellent 20.

Comment 20:

Did you perform an inter- and intraexaminer validation of the goniometric assessment of shoulder ROM?

Response 20: Thank for your comment! No, but there were article concerning.

Riddle and colleagues revealed that intraclass correlation coefficients for intratester reliability were .98 for flexion, .98 for abduction, .94 for extension, .90 for horizontal adduction, .99 for lateral rotation, and .94 for medial rotation. ICCs reflecting intertester reliability were notably lower, ranging from .26 to .90. All measurements in this study were done by the same occupation therapist.

Comment 21:

According to your data the restriction of shoulder ROM was mild. Do you think that inclusion of chronic patients with frozen shoulder and greater stiffness could have affected your results?

Response 21: Thank for your comment! We put it in the limitation of this study.

Third, we did not conduct subgroup analyses based on the stage of the disease (e.g., freezing, frozen, and thawing stages), which may have influenced the treatment outcomes. Future research should include patients at various disease stages and follow them for longer periods to allow for a more comprehensive assessment. Finally, conclusions regarding the optimal type or duration of rehabilitative intervention could not be drawn from this study.

Comment 22:

Response 22: Thank for your comment! We removed the references and re-organized this paragraph.

Line 200. Reference #21 is not relevant

Response: we removed this reference.

Line 203. restoration of flexion was impeded. This is not correct since the disease is chronic and resistant to treatment.

several literatures. Please improve

Response: Thank you! We modified the sentence.

Interestingly, in the group combining corticosteroids injection and rehabilitation, the restoration of flexion was impeded and a second injection was warranted. Meanwhile, abduction ROM failed to increase at the second and third endpoints. Futher study should be conducted to find the pathology which might impede the ROM in the second and third injection.

lines 224-226. The references are not relevant.

Response: We removed this reference.

Lines 228-261. The paragraph is not relevant to the study. It is preferable to describe the various mobilization protocols in shoulder stiffness.

Response: We deleted this paragraph.

Comment 23:

Extensive editing is required.

Response 23: Thank for your comment! The English editing had completed.

Reviewer 3 Report

Comments and Suggestions for Authors

The manuscript entitled "The effect of Novel Ultrasound-guided Subacromial Subdeltoid Bursa and Coracoid Humeral Ligament Corticosteroid Injection in Treating Frozen Shoulder" aims to investigate the effect of novel ultrasound-guided steroid injection of subacromial subdeltoid bursa (SSB) and coracoid humeral ligament (CHL) in treating frozen shoulder, and the additional effect of regular electrotherapy and traditional rehabilitation exercise. The following comments needs to be addressed.

1. The introduction refers to old literature. interestingly there are little references after 2020. MOst of the works referred are old. The authors are advised to refer the recent publications (post 2020) in the relavant area to contribute tangible advancement in the field.

2. Include references form 2023 & 2024.

3. The introduction should be modified significantely.

4. The authors should justify the rationale for use of the chosen statistical methods (Chi square & T test). in the current study there are more than 2 categories of variables, and chi square test is difficult to interpret when there are more than 2 variables. 

5. What care is taken to avoid type 1 errors in the t-test. 

6. Conclusion is totally missing. please include.

Comments on the Quality of English Language

Moderate editing required.

Author Response

Comment 1:

The introduction refers to old literature. interestingly there are little references after 2020. Most of the works referred are old. The authors are advised to refer the recent publications (post 2020) in the relavant area to contribute tangible advancement in the field.

Response 1: Thank for your comment! We add the recent articles in the revised manuscript including references from 2023 & 2024.We re-wrote the introduction with recent reference.

Shoulder pain is the third most frequently encontered musculoskeletal complaint in clinics and imposes a significant burden on patients' daily activities as well as healthcare costs 1. Adhesive capsulitis is a common shoulder disorder with a lifetime prevalence of 2–5%, particularly affecting middle-aged women 2, 3. Commonly referred to as "frozen shoulder," the condition is characterized by pain and progressive, global glenohumeral joint range of motion (ROM) limitation 3. Although adhesive capsulitis is generally considered a self-limiting disease that resolves within two to three years, symptoms can be debilitating and may persist in up to 50% of patient 4.

Pathogenesis of adhesive capsulitis is complex and likely multifactorial. Chronic inflammation, fibroblast proliferation, and an imbalance in extracellular matrix turnover ultimately lead to capsular stiffness 4. Notably, the predominant pathological changes were found at the rotator interval, both radiologically and histologically 5-7. The rotator interval is a triangular anatomical space located at the anterosuperior aspect of the shoulder, bordered superiorly by the supraspinatus, inferiorly by the subscapularis, and with the coracoid process forming its base 8. Key structures within this interval include the coracohumeral ligament (CHL), the superior glenohumeral ligament (SGHL), the long head of the bicep tendon, and the anterior joint capsule. These components play a critical role in maintaining the stability of the glenohumeral joint. Contracture of the rotator interval along with fibrosis, hyalinization, and fibroid degeneration of the CHL has been linked to the loss of shoulder external rotation in adhesive capsulitis 5, 9. On the other hand, studies had found elevated levels of inflammatory mediators not only in the joint capsule but also in the subacromial-subdeltoid bursa (SSB) 10-13.

There is a wide variety of treatments is available for adhesive capsulitis, ranging from non-operative approaches such as nonsteroidal anti-inflammatory drugs (NSAIDs), physiotherapy, corticosteroid injections, and joint hydrodilatation, to surgical options like manipulation under anesthesia and arthroscopic capsular release 14.  Corticosteroids exert a general suppressive effect on the inflammatory response and inhibit the differentiation of fibroblasts into myofibroblasts. Given that both the SSB and the CHL play a role in the development of adhesive capsulitis—with the CHL primarily contributing to the loss of external rotation—we proposed an extra-articular approach. These two sites can be efficiently and accurately targeted under ultrasound guidance through the same puncture site.

Comment 2:

Include references form 2023 & 2024.

Response 2: Thank for your comment! We add the recent articles in the revised manuscript including references from 2023 & 2024.

Comment 3:

The introduction should be modified significantely.

Response 3: Thank for your comment! We re-wrote the introduction.

Comment 4:

The authors should justify the rationale for use of the chosen statistical methods (Chi square & T test). in the current study there are more than 2 categories of variables, and chi square test is difficult to interpret when there are more than 2 variables. 

Response 4: Thank for your comment! Statistics had been conducted once again.

Comment 5:

What care is taken to avoid type 1 errors in the t-test. 

Response 5: Thank for your comment! Statistics had been conducted once again.

Two sample -t-test was used to compare differences of age, height, weight, BMI, baseline ROM and SPADI between Group 1 and Group 2. And Fisher’s exact test was used to compare difference of gender distributions between the 2 groups.

Comparisons of ROM and SAPDI across over all time points between the 2 groups were assessed by mixed effects model with repeated measurements (MMRM) analysis. The MMRM analysis was based on unstructured variance-covariance matrix and de-nominator degree of freedom using Kenward Roger method including a between-group fixed effect (Group 1 vs. Group 2) and a within-subject repeated factor (evaluation time: pre-treatment, 4 weeks after treatment, 8 weeks after treatment, and 12 weeks after treatment), and group-by-time interaction with individual subject as a cluster unit. Additionally, for assessment of ROM, the MMRM model also included the ROM assessment from the untreated (normal) arm and time-by- ROM assessment from untreated (normal) arm as covariates.

All significance levels were set at two-sided α < 0.05, and SAS Studio version 3.1M1 was used to perform all statistical analyses.

This is a preliminary proof of concept study as such the sample size is not powered. Further enrollment will be continued and the study will be powered for confirmatory purpose.

Comment 6:

Conclusion is totally missing. please include.

Response 6: Thank for your comment! We add conclusion in the revised manuscript.

Conclusions

A dual-target, ultrasound-guided corticosteroids injection to the SSB and CHL was effective in decreasing shoulder pain and disability in patients with adhesive capsulitis. Significant improvements in shoulder ROM were also achieved. Therefore, this approach may serve as an alternative treatment option for affected patients.

Reviewer 4 Report

Comments and Suggestions for Authors

This study aims at the value of additional physiotherapy, not the effect of ultrasound guided steroid injections. This fact should be clearly stated and considered in all parts of the manuscript.

Therefore in the discussion: The main findings of this study is that physiotherapy did not further improve  the effect of the first ultrasound-guided 197 steroid injection of SSB and CHL. Steroid injections alone may be more cost-effective than steroid plus physiotherapy or physiotherapy alone [32]. This should be clarified in the abstract, the introduction, the results and the discussion sections.

The sentence, that "from our results, single ultrasound-guided steroid injection of SSB and CHL resulted in satisfying pain reduction, functional improvement and ROM restoration in flexion, abduction and external rotation" should be omitted as it may be misleading. This study does not address this issue.

Comments on the Quality of English Language

No comments.

Author Response

Comment 1:

This study aims at the value of additional physiotherapy, not the effect of ultrasound guided steroid injections. This fact should be clearly stated and considered in all parts of the manuscript.

Response 1: Thank for your comment! We Deleted this sentence

However, no additional benefits of electrotherapy and traditional rehabilitation exercises were observed.

Comment 2:

Therefore in the discussion: The main findings of this study is that physiotherapy did not further improve the effect of the first ultrasound-guided 197 steroid injection of SSB and CHL. Steroid injections alone may be more cost-effective than steroid plus physiotherapy or physiotherapy alone [32]. This should be clarified in the abstract, the introduction, the results and the discussion sections.

Response 2: Thank for your comment! We put it in the limitation of this study There were several limitations in this study. First, the sample size was relatively small, which may limit the generalizability of the findings. Second, the follow-up period was restricted to only three months, which may not have been sufficient to fully capture the long-term effects of the treatment or potential relapses. Third, we did not conduct subgroup analyses based on the stage of the disease (e.g., freezing, frozen, and thawing stages), which may have influenced the treatment outcomes. Future research should include patients at various disease stages and follow them for longer periods to allow for a more comprehensive assessment. Finally, conclusions regarding the optimal type or duration of rehabilitative intervention could not be drawn from this study. It remains unclear whether certain rehabilitation protocols are more effective than others when combined with injection therapy. Further studies are necessary to identify the most effective rehabilitation strategies to maximize patient recovery.

Comment 3:

The sentence, that "from our results, single ultrasound-guided steroid injection of SSB and CHL resulted in satisfying pain reduction, functional improvement and ROM restoration in flexion, abduction and external rotation" should be omitted as it may be misleading. This study does not address this issue.

Response 3: Thank for your comment! We deleted the word” satisfying”.

Round 2

Reviewer 1 Report

Comments and Suggestions for Authors

Congratulations to the authors for addressing the comments raised int he previous round of review

Comments on the Quality of English Language

Minor language polishing could be done

Author Response

We sincerely appreciate the reviewer’s insightful comments, which have greatly contributed to improving the quality of our manuscript. We have also addressed a few grammatical errors and refined the English writing.

Reviewer 3 Report

Comments and Suggestions for Authors

The authors have addressed the comments, and the manuscript may now be accepted.

Comments on the Quality of English Language

Minor editing required.

Author Response

We express our heartfelt gratitude for the reviewer’s valuable critiques, which have significantly enhanced the quality of our manuscript. We have addressed a few grammatical errors and refined the English writing for better readability.

Reviewer 4 Report

Comments and Suggestions for Authors

Again, the statement that "ROM in 289 the frontal, sagittal and coronal planes were all improved with a single percutaneous ul- 290 trasound-guided injection to the SSB and CHL" cannot be done. There is no comparative groups for this outcome of the study. The first sentence has to be that there is no effect of additional physiotherapy. Please clarify in all parts of the mansucript.

Comments on the Quality of English Language

Has to be improved (eg line 61)

Author Response

Comment: Again, the statement that "ROM in 289 the frontal, sagittal and coronal planes were all improved with a single percutaneous ul- 290 trasound-guided injection to the SSB and CHL" cannot be done. There is no comparative groups for this outcome of the study. The first sentence has to be that there is no effect of additional physiotherapy. Please clarify in all parts of the mansucript.

Reply: We sincerely thank the reviewer for their valuable critique. We fully agree that the main finding of this study is that adding physiotherapy to the injection approach did not result in significantly greater benefits. Accordingly, we have made revisions throughout the manuscript to ensure clarity and alignment with this key point.

The background is now revised as: “The objective of this study was to investigate the effect of ultrasound-guided corticosteroid injection to the subacromial subdeltoid bursa (SSB) and coracoid humeral ligament (CHL) in treating adhesive capsulitis, with a particular focus on evaluating the potential benefits of regular electrotherapy and conventional rehabilitation exercises.” And the first sentence of the results has been updated to “Electrotherapy and traditional rehabilitation exercises did not enhance the effectiveness of this injection approach.” The conclusion was revised as “Physiotherapy did not offer additional benefits when combined with ultrasound-guided cortico-steroid injection to the CHL and SSB. The injection alone significantly improved pain, disability, and ROM in patients with adhesive capsulitis. Further research is required to determine the optimal type, duration, and cost-effectiveness of rehabilitation programs for managing this condition.”

Similarly, in the Introduction, we made the following change “The objective of this study was to assess the effectiveness of SSB and CHL cortico-steroids injection in treating adhesive capsulitis, with a specific emphasis on examining the potential additional benefits of electrotherapy and conventional rehabilitation exercises.”

The First paragraph of the Discussion was revised as “The main finding of this study was that physiotherapy did not enhance the effectiveness of a single percutaneous ultrasound-guided injection to the SSB and CHL in treating adhesive capsulitis. This injection approach resulted in significant improvements in pain, upper extremity function, and ROM across the frontal, sagittal, and coronal planes.” Additionally, in line with the reviewer’s suggestion, we avoided language that might mislead readers. For example, we revised statements such as “From our results, single ultrasound-guided steroid injection of SSB and CHL resulted in satisfying pain reduction, functional improvement, and ROM restoration in flexion, abduction, and external rotation.” to “Pain reduction, functional improvement and restoration in flexion, abduction and external rotation ROM were observed after a single ultrasound-guided steroid injection to the SSB and CHL.” We added “Nevertheless, our results emphasize the promise and cost-effectiveness of using extra-articular corticosteroid injections as a standalone treatment option.” to the part of the Discussion reviewing the benefits of physiotherapy for emphasis.

The Conclusion was revised to “Physiotherapy did not provide additional benefits for patients with adhesive capsulitis beyond those achieved with a dual-target, ultrasound-guided corticosteroid injection targeting the SSB and CHL. The injection alone was effective in decreasing shoul-der pain and disability.”

Round 3

Reviewer 4 Report

Comments and Suggestions for Authors

We further addressed the issue of the necessity of repeatedly injections.

Indeed, the manuscript improved with this revision. However, I still recommend some additional minor changes:

Line 34: I recommend „required to further optimize current physiotherapy with electrotherapy and traditional rehabilitation exercises after ultrasound-guided corticosteroid injections“ instead of „required to determine the optimal type, … condition“. 

Line 289: I recommend to omit  "We further addressed the issue of the necessity of repeatedly injections.“ has to be omitted.

Line 362: I recommend to omit „Significant improvements in shoulder ROM were also  … warrants further evaluation“. Instead I propose: "Because of the small sample size of this study, a larger study to optimize current physiotherapy with electrotherapy and traditional rehabilitation exercises after ultrasound-guided corticosteroid injections is warranted."

Title: Accordingly, the title should be changed to: „Electrotherapy and traditional rehabilitation exercises lack effectiveness after ultrasound-guided corticosteroid injections in adhesive capsulitis“. You may add: "- a pilot study" or change the title to "„Electrotherapy and traditional rehabilitation exercises may lack effectiveness after ultrasound-guided corticosteroid injections in adhesive capsulitis“

  Comments on the Quality of English Language

n.a.

Author Response

Comment 1:

Line 34: I recommend „required to further optimize current physiotherapy with electrotherapy and traditional rehabilitation exercises after ultrasound-guided corticosteroid injections“ instead of „required to determine the optimal type, … condition“. 

Response 1:

First, we must thank the reviewer for the positive feedback. We have revised the statement to “Further research is required to optimize current physiotherapy with electrotherapy and traditional rehabilitation exercises after ultrasound-guided corticosteroid injections.”

Comment 2:

Line 289: I recommend to omit  "We further addressed the issue of the necessity of repeatedly injections.“ has to be omitted.

Response 2:

The line was omitted.

Comment 3:

Line 362: I recommend to omit „Significant improvements in shoulder ROM were also  … warrants further evaluation“. Instead I propose: "Because of the small sample size of this study, a larger study to optimize current physiotherapy with electrotherapy and traditional rehabilitation exercises after ultrasound-guided corticosteroid injections is warranted."

Response 3:

The line was modified as "Because of the small sample size of this study, larger-scale research to optimize current physiotherapy with electrotherapy and traditional rehabilitation exercises after ultrasound-guided corticosteroid injections is warranted."

Comment 4:

Title: Accordingly, the title should be changed to: „Electrotherapy and traditional rehabilitation exercises lack effectiveness after ultrasound-guided corticosteroid injections in adhesive capsulitis“. You may add: "- a pilot study" or change the title to "„Electrotherapy and traditional rehabilitation exercises may lack effectiveness after ultrasound-guided corticosteroid injections in adhesive capsulitis“.

Response 4:

We thank the reviewer for this suggestion. The title has been changed to “Electrotherapy and traditional rehabilitation exercises may lack effectiveness after ultrasound-guided corticosteroid injections in adhesive capsulitis.”